# Isolation, Characterization and Anticancer Activity of Two Bioactive Compounds from *Arisaema flavum* (Forssk.) Schott

**DOI:** 10.3390/molecules27227932

**Published:** 2022-11-16

**Authors:** Sobia Nisa, Yamin Bibi, Saadia Masood, Ashraf Ali, Sadia Alam, Maimoona Sabir, Abdul Qayyum, Waqas Ahmed, Sarah Alharthi, Eman Y. Santali, Saif A. Alharthy, Waleed M. Bawazir, Majed N. Almashjary

**Affiliations:** 1Department of Microbiology, The University of Haripur, Haripur 22620, Pakistan; 2Department of Botany, PMAS-Arid Agriculture University Rawalpindi, Rawalpindi 46300, Pakistan; 3Department of Statistics & Mathematics, PMAS-Arid Agriculture University Rawalpindi, Rawalpindi 46300, Pakistan; 4Department of Chemistry, The University of Haripur, Haripur 22620, Pakistan; 5Department of Agronomy, The University of Haripur, Haripur 22620, Pakistan; 6Department of Chemistry, College of Science, Taif University, Taif 21944, Saudi Arabia; 7Center of Advanced Research in Science and Technology, Taif University, Taif 21944, Saudi Arabia; 8Department of Pharmaceutical Chemistry, College of Pharmacy, Taif University, Taif 21944, Saudi Arabia; 9Department of Medical Laboratory Sciences, Faculty of Applied Medical Sciences, King Abdulaziz University, Jeddah 21589, Saudi Arabia; 10Toxicology and Forensic Sciences Unit, King Fahd Medical Research Center, King Abdulaziz University, Jeddah 21589, Saudi Arabia; 11Hematology Research Unit, King Fahd Medical Research Center, King Abdulaziz University, Jeddah 21589, Saudi Arabia; 12Animal House Unit, King Fahd Medical Research Center, King Abdulaziz University, Jeddah 21589, Saudi Arabia

**Keywords:** MTT assay, medicinal plants, isolation and purification, breast cancer, *Arisaema flavum* (Forssk.)

## Abstract

Medicinal plants play important role in the public health sector worldwide. Natural products from medicinal plants are sources of unlimited opportunities for new drug leads because of their unique chemical diversity. Researchers have focused on exploring herbal products as potential sources for the treatment of cancer, cardiac and infectious diseases. *Arisaema flavum* (Forssk.) is an important medicinal plant found in the northwest Himalayan regions of Pakistan. It is a poisonous plant and is used as a remedy against snake bites and scorpion stings. In this study, two bioactive compounds were isolated from *Arisaema flavum* (Forssk.) and their anticancer activity was evaluated against human breast cancer cell line MCF-7 using an MTT assay. The crude extract of *Arisaema flavum* (Forssk.) was subjected to fractionation using different organic solvents in increasing order of polarity. The fraction indicating maximum activity was then taken for isolation of bioactive compounds using various chromatographic and spectroscopic techniques such as column chromatography, thin-layer chromatography (TLC), gas chromatography–mass spectrometry (GC-MS), Fourier transform infrared spectroscopy (FTIR) and nuclear magnetic resonance spectroscopy (NMR). Crude extract of *Arisaema flavum* (Forssk.), as well as various fractions extracted in different solvents such as n-hexane, chloroform and ethyl acetate, were tested against human breast cancer cell line MCF-7 using an MTT assay. The crude extract exhibited significant dose-dependent anticancer activity with a maximum activity of 78.6% at 500 µg/mL concentration. Two compounds, hexadecanoic acid ethyl ester with molecular formula C_18_H_36_O_7_ and molar mass 284 and 5-Oxo-19 propyl-docosanoic acid methyl ester with molecular formula C_26_H_50_O_3_ and molecular mass 410, were isolated from chloroform fraction. These compounds were tested against the MCF-7cell line for cytotoxic activity and exhibited a significant (*p* < 0.00l) decrease in cell numbers for MCF-7 cells with IC_50_ of 25 µM after 48 h of treatment. Results indicated that *Arisaema flavum* (Forssk.) possesses compounds with cytotoxic activity that can further be exploited to develop anticancer formulations.

## 1. Introduction

In recent years, natural products have experienced a renaissance in drug discovery, mainly due to their superior chemical diversity over synthetic compounds and their drug-like properties [1]. About 80% of the world’s population is dependent on herbal remedies in one way or another [2]. Plant-derived pharmaceuticals have comprised a significant market share of prescribed medicines all over the world. The ethno-botanical importance of plants serves as a guide to explore them for the isolation of bioactive compounds that can be developed into drugs [3]. In addition to chemical compounds being used as drugs, some chemical entities from plants can also serve as building blocks for bioactive drugs [4]. These bioactive ingredients can be utilized in the synthesis of bioactive compounds with enhanced activity and applications. Etoposide and teniposide are well-known examples of semi-synthetic drugs that are commonly used to treat skin cancer [5]. The precursor of these compounds is epipodophyllotoxin, which was isolated from the May apple *Podophyllum* sp. [6].

*Arisaema flavum* (Forssk.) belongs to the family Araceae, which grows on mountain slopes, farmland and roadsides of hilly areas at altitudes of 3500–4300 m [7]. In Pakistan, it is found in the Kashmir and Kaghan areas [8]. It is a poisonous plant, ethno-botanically used as a remedy against scorpion stings and snake bites [9]. It is also known for the treatment of chronic tracheitis, epilepsy and tetanus. Boiled roots of *Arisaema flavum* (Forssk.) are used to treat stomach pain. The root extract of the plant is also known for its insecticidal and anthelmintic properties [10]. Antibacterial properties of *Arisaema flavum* (Forssk.) have also been reported in [11]. A novel lectin, known as *Arisaema flavum* (Forssk.) lectin, was isolated from rhizomes of *Arisaema flavum* (Forssk.) and exhibited significant anti-proliferative activity against J774 and P388D1 cancer cell lines [12].

The main challenge in phyto-medicine is the separation and purification of the active ingredient from the matrices present in extracts. Plant extracts contain various types of bioactive compounds or phytochemicals in addition to some unwanted analytes and their separation is a big challenge. These unwanted matrices may act synergistically and minimize the beneficial effect of bioactive compounds. In the present study, two bioactive compounds were isolated from the *Arisaema flavum* (Forssk.) rhizome extract, and their anticancer activities against breast cancer cell lines were checked. The isolation and purification of bioactive compounds were carried out systematically using solvent extraction, column chromatography and thin-layer chromatography followed by gas chromatography–mass spectrometry (GC-MS) analysis. The isolated compounds were characterized by nuclear magnetic resonance (NMR) spectroscopy and GC-MS analysis, and their structures were elucidated. The anticancer activities of these bioactive compounds and crude extract were checked against breast cancer cell lines.

## 2. Experimental

### 2.1. Plant Collection

*Arisaema flavum* (Forssk.) was collected from Kaghan valley located in Khyber Pakhtunkhwa, Pakistan, and identified by Prof. Dr. Mir Ajab Khan, Department of Plant Sciences, Quaid-i-Azam University, Islamabad, Pakistan. The sample specimen was deposited at the herbarium of the Department of Plant Sciences, Quaid-i-Azam University, Islamabad, Pakistan, to keep on record.

### 2.2. Extraction

Rhizomes of *Arisaema flavum* (Forssk.) were dried and ground to a fine powder. Ground powder (2.5 kg) was macerated with methanol (2500 mL) in closed glass container for two weeks at room temperature. Methanol extract was filtered using vacuum filter and filtration was repeated twice to obtain maximum yield. The methanol filtrate was concentrated by using rotary evaporator at 40 °C. As a result, 240 g brownish semisolid residue was obtained).

### 2.3. Fractionation

Crude methanol extract of *Arisaema flavum* (Forssk.) (240 mL) was subjected to fractionation by suspending it in 200 mL distilled water and then partitioned with different solvents of increasing order of polarity such asn-hexane, chloroform, ethyl acetate and methanol using separating funnel. All fractions, including aqueous fraction, were dried using rotary evaporator. Fractions so obtained were hexane (2 g), chloroform (24 g), ethyl acetate (26 g), methanol (32 g) and aqueous (38 g).

### 2.4. Anticancer Activity of Arisaema flavum (Forssk.) Crude Extract and Fractions

Breast cancer cell line MCF-7 (human breast estrogen-dependent adenocarcinoma) was purchased from LGC standards, Teddington, UK. Cells culture was maintained in Dulbecco’s modified Eagle’s medium (DMEM) supplemented with 15% fetal bovine serum (FBS), 40 µg/mL gentamycin, 100 units/mL penicillin and 1.04 mg/mL streptomycin.

Anticancer activity of crude extract and different fractions of *Arisaema flavum* (Forssk.) was determined by MTT assay. MCF 7 cells were seeded in 96-well plates at the density of 5000 cells/well in 100 μL medium (RPMI 1640) and incubated overnight in 5% CO_2_ at 37 °C_._ After incubation, various concentrations (10, 25, 50, 100, 200, 300, 400 and 500 µg/mL) of *Arisaema flavum* (Forssk.) crude extracts and fractions (200 µg/mL) were added in microwell plates, and incubation was performed for further 24 h.

After incubation, 10 μL MTT reagent was added to each well, and plates were additionally incubated for 4 h. Then, 100 μL of DMSO solution was added to the wells to dissolve the MTT crystals. The plates were incubated overnight at 37 °C. The plates were read for optical density at 570 nm, using a microplate reader.

Results were expressed as percentage inhibition depending upon the average absorbance values of samples in comparison to average absorbance values of blank media and media with cells.

The percent inhibition was calculated using Equation (1).
(1)$$\%\mathrm{inhibition}=\frac{\left(B-A\right)-\left(C-A\right)\;}{\left(B-A\right)}\times 100\;$$
where A = Average absorbance of media, B = Average absorbance of media with cells, C = Average absorbance of extract sample

Based on percentage inhibition caused by crude extracts IC_50_ value was calculated as 180.25 µg/mL, and the nearest concentration of 200 µg/mL was selected to test anticancer activity of fractions.

### 2.5. Isolation of Compounds from Chloroform Fraction

#### 2.5.1. Chromatographic Analysis

Silica gel 60 (Sigma–Aldrich) with pore size 0.035–0.070 mm was used for column chromatography. Thin-layer chromatography was carried out on TLC Plates, silica gel 60 F254 with fluorescent indicator, and pre-coated aluminum cards (0.2 mm thickness) from Merck (Germany).

Chloroform fraction (24 g) was loaded on a column and stepwise elution was carried out first with hexane and then with various concentrations of the organic solvents, chloroform, ethyl acetate and finally with methanol. Fractions indicating the presence of major compound (in the form of oily droplets) during TLC were combined and preceded for GCMS where single peak indicated the presence of pure compound-I.

Another compound was also isolated over silica gel using mobile phase chloroform: ethyl acetate 10:90. First TLC and then GCMS was performed to confirm the purity of compound-II.

#### 2.5.2. GC-MS Analysis of Isolated Compounds

GC (Agilent Technologies) 5890 with Helium gas as the carrier was used along with electron spray ionization connected to auto sampler injection system. Acquired Method Default was used for analysis with temperature range from 40 to 340 °C with a run of 28 min.

#### 2.5.3. FTIR Analysis of Isolated Compounds

Thermo Nicolet 380 FT-IR Spectrophotometer was used for analysis of compounds. It was operated using OMINIC version 7.3 controls and processing software from Thermo Electron Corporation, using Sodium Chloride discs. Absorption bands were coated in wave numbers (cm^−1^). Isolated compounds were dissolved in chloroform and mixed with nujol mull for their analysis.

#### 2.5.4. NMR Analysis of Isolated Compounds

JEOL Eclips 400 NMR Spectrometer with JeoL Delta Version 7.2 control and processing software was used to perform different NMR experiments. Peak positions were quoted on the δ scale in comparison to an internal standard. Dried pure compounds were weighed (5–10 mg) and dissolved in (2 mL) deuterated chloroform (Merck) for NMR. Proton, Carbon, DEPT experiments were run on NMR.

### 2.6. Anticancer Activity of Isolated Compounds

Isolated compounds (I and II) were subjected to MTT assay for cytotoxicity analysis using an MCF-7 cell line. Different concentrations (0.78, 1.56, 3.12, 6.25, 12.5, 25, 50, 100 µM) of pure isolated compounds were tested against MCF-7 cell line. Final volume of 200 µL was prepared for each concentration with 0.01% DMSO. Tamoxifen (5 µM), Actinomycin-D (4 µM) and Anastrozole (5 µM) were used as positive control at concentrations as reported by Kakrani et al. [13]. Observations were recorded after 48 h of incubation. Experiments were replicated in triplicates and percentage inhibition of cancer cells was recorded for treated and untreated cells, using formula as previously described in Section 2.4.

### 2.7. Statistical Analysis

All the assays were performed in triplicate. Results of crude extract activity were calculated for average percentage inhibition while data obtained from activity of fractions were statistically analyzed by ANOVA and LSD using MSTATC.

## 3. Results

### 3.1. Anticancer Activity of Arisaema flavum (Forssk.)

The results of anticancer activity are expressed as percent inhibition (Figure 1A). The crude extract of *Arisaema flavum* (Forssk.) rhizome indicated a dose-dependent response with 78.6% inhibition at 500 µg/mL with an average absorbance of 0.084 (Figure 1A). The best inhibition activity (83%) was exhibited by the methanol fraction against the MCF-7 cell line followed by chloroform (76%), ethyl acetate (39.5%), hexane (25%) and aqueous (1.2%) fractions, as indicated in Figure 1B.

### 3.2. Isolation and Characterization of Bioactive Compounds from Arisaema flavum (Forssk.) Extract

The chloroform fraction of *Arisaema flavum* (Forssk.) was purified by column chromatography followed by GC-MS analysis. Two compounds were isolated from the chloroform fraction of *Arisaema flavum* (Forssk.), in which compound-I was yellow in color with oily nature.

#### 3.2.1. Spectroscopic Analysis of Isolated Compounds

The spectroscopic analysis of compound-I and compound-II are described in the Appendix A.

#### 3.2.2. Chromatographic Analysis of Isolated Compounds

The single peak on the chromatogram in Figure 2 indicates the presence of a pure compound with a retention time of 14.8 min. The m/z ion peak at 285 in the mass spectrum indicated that the molecular mass of the compound is 284 with a base peak at 88, as shown in Appendix A. Fragment ions at m/z 269, 255, 241, 227, 213, 199, 185, 171, 157, 143, 129, 115, and 101 represent the loss of a methyl (-CH2) group from molecular ion [M+H-14f] from each respective daughter ion. A fragment ion at *m*/*z* 88 with maximum abundance (600,000) suggested the fragment of an ethyl acetate ion (Appendix A. Compound-I was named hexadecanoic acid ethyl ester with molecular formula C_18_H_36_O_2_ (Appendix A).

Similarly, compound-II was identified as 5-Oxo-19 propyl-docosanoic acid methyl ester, which was also oily in appearance with a yellow color. The structure of the compound revealed it is also ester in nature. GC analysis of compound-II showed the presence of a single peak at 11.5 min that corresponds to the presence of pure compound (Figure 2B). The MS analysis of compound-II shows an m/z ion peak at 410 with a base peak at 88. Other peaks were observed at 55, 101, 129, 157, 185.9, 241, 255 and 284, as indicated in Appendix A.

#### 3.2.3. FT-IR Analysis of Isolated Compounds

FTIR spectral analysis was carried out for compound-I in chloroform indicating the presence of stretchy peaks at 2922.8 and at 2851.29 cm^−1^ corresponding to the presence of alkane (Figure 3A). A peak was observed at 1738.88 cm^−1^ indicating the presence of ester linkage C = O (Figure 3A). A stretch was also observed at 1179.08 cm^−1^, which corresponds to the presence of C-O in esters.

Similarly, the FT-IR spectrum of compound-II has different peaks as shown in Figure 3B. Absorption peaks at 2924, 2853, 1463 and 1376 and bending at 800 and 722 cm^−1^ all corresponded to the presence of an aliphatic chain in the compound. Absorption bands at 1736 cm^−1^ and at 1171, 1108 and 1054 indicated the presence of functional groups (C = O) and (C-O) representing vibrations of ester molecules (Figure 3B).

#### 3.2.4. ^1^H-NMR of Isolated Compounds

^1^H-NMR of compound-I indicated the presence of 36 protons while the carbon NMR indicated the presence of 18 carbons (Appendix A). DEPT spectra revealed that the compound contains one quaternary carbon at the δ179.3 position. The presence of carbon at this position co-relates its linkage with oxygen. Carbon DEPT further confirmed the presence of 2 CH_3_ and 15 CH_2_. The signal position for carbons indicated the presence of an aliphatic chain in the compound with ester linkage (Table 1A and Appendix A).

In Appendix A, ^1^H-NMR indicated 36 protons, indicating signals for two protons at 4.11 (2H, dd, *J =* 7.14 Hz, H). Triplet signals were observed at 2.28 (2H,t,*J* = 7.60 Hz, H), 1.59 (2H,t,*J =* 7.23 Hz, H) and 0.88 (3H,t,*J* = 6.6 Hz, H16) while multiplet signals were spotted at 1.25 (5H, m, H). A broad singlet was observed at 1.27 (22 H,s). C^13^ NMR indicated the presence of 18 carbons with 2 methyl and 15 methylene groups_._ Signals at position δ 173.9 for (C = O) and δ 60.15 for (CH_2_-O) corresponded to the presence of ester. Signals at *δ* 1.00–0.67 indicated the presence of saturated carbons [14,15]. By a comparison of the above-stated results with published data and by interpreting the above results, it was concluded that the isolated compound was hexadecanoic acid ethyl ester (Appendix A). The proposed structure of compound-I on the basis of characterization has been elucidated in Appendix A.

The proton NMR of compound-II indicated the presence of 50 protons, among them three singlets, two multiplets and two doublets of doublets were recorded (Appendix A). A broad multiplet at δ_H_1.25 describes the integration of the 26 protons of the aliphatic alkyl chain consisting of eleven methylene groups from C-8 to C-18 (δc 22.7–32.0) and two methylene groups at C-20 and C-23 (δc 29.7). Carbon spectra indicated the presence of 26 carbons, while DEPT spectra revealed that the compound contains two quaternary carbons at δ188.45 and δ174.45. The presence of carbons peaks at these positions indicated its bonding with oxygen, which may be due to ester linkage. Further, the presence of three CH_3,_ 1CH and 20 CH_2_ groups in the compound was confirmed by DEPT spectra. The presence of aliphatic chains was also observed on basis of signals (Table 1B and Appendix A). All these results suggest that the isolated compound is 5oxo-19-propyl-docosanoic acid methyl ester (Appendix A).

### 3.3. Cytotoxic Activity of Isolated Compounds

Results of cytotoxic activity indicated that both compounds were able to effectively control the growth of the MCF-7 cancer cell line; however, docosanoic acid methyl ester was significantly more potent with 80% inhibition at a concentration of 100 µM in comparison to the 67% inhibition caused by hexadecanoic acid ethyl ester at the same concentration (Figure 4A,B).

## 4. Discussion

Medicinal plants have been known for their therapeutic effects in many areas including antimicrobial, anti-inflammatory, and for cardiovascular diseases and as anticancer agents [16]. Based on folk use and some earlier reports, *Arisaema flavum* (Forssk.) Schott rhizome crude extract and fractions were subjected to tests against the MCF 7 cancer cell line. The results indicated a dose-dependent response of the breast cancer cell line with an IC _50_ value of 200 µg/mL. A similar dose-dependent response of *Debregeasia salicifolia* Rendle (DS) was also reported against the MCF 7 cell line [17]. Based on the IC _50_ value of crude extract, a 200 µg/mL concentration of fractions was selected for further processing. Appendix A shows 50% inhibition of crude extract of *Arisaema flavum* (Forssk.) at 200 µg/mL. The highest anticancer activity was observed for the methanol fraction indicating the polar nature of active components, while moderately polar active compounds were also present in the chloroform fraction as revealed from the activity of the chloroform fraction [18]. Vijayarathna and Sasidharan also reported the best cytotoxic activity of the methanol extract of *Elaeis guineensis* Jacq. against the MCF-7 cell line [19].

The chloroform extract was taken for the isolation of bioactive compounds following repetitive column and thin-layer chromatography leading to the isolation of two bioactive compounds, hexadecanoic acid ethyl ester and 5oxo-19-propyl-docosanoic acid methyl ester, which were characterized initially by GC-MS analysis and later by FTIR and NMR analysis. The present study, in line with the previous findings, highlighted the significance of the use of a simple column technique for the purification of compounds from plant extracts [20].

FT-IR has proven to be a valuable tool for the characterization and identification of compounds or functional groups (chemical bonds) present in an unknown mixture of plant extract [14,15]. Compound-I indicated stretchy peaks at 2922.8 and 2851.29 cm^−1^ corresponding to the presence of alkane and a peak at 1738.88 cm^−1^ indicated the presence of ester linkage C = O (Figure 3A). A stretch was also observed at 1179.08 cm^−1^, which corresponds to the presence of C-O in esters. Similar peaks were also observed by Kaliyaperumal et al. [21] who isolated Palmitic acid, 2- (tetradecyloxy) ethyl ester from methanolic extract of *Bauhinia purpurea* L. Compound-II indicated peaks at 1461 and 1372 cm^−1^ relating to the presence of CH_3_ in aliphatic compounds. Results of ^1^H-NMR indicated the presence of 36 protons in compound-I and 50 protons in compound-II. The results of C-NMR were in accordance with Anu and Rao [22] who also reported a broad singlet at the same position. The aliphatic straight chain mostly shows these signals. The triplet at *δ* 0.85 is specific for the terminal –CH_3_ group.

This is the first report on the isolation of hexadecanoic acid ethyl ester from *Arisaema flavum* (Forssk.) as previously isolated from marine bacteria, and its antibacterial potential was investigated. Fatty acids and long-chain esters are associated with each other and have been reported as cell membrane components in both animals and plants [23]. Oligosaccharides and their esters are a significant group of phytochemical compounds. They are reported from roots, rhizomes, stems, barks, leaves, aerial, and whole parts of medicinal plants. Principally, they are energy-storing compounds but can also play role in the treatment of diseases [24,25,26]. Our study reports the dose-dependent effects of both compounds on the MCF-7 cell line; however, docosanoic acid methyl ester is significantly more potent with 80% inhibition at a concentration of 100 µM in comparison to only 67% inhibition caused by hexadecanoic acid ethyl ester. Similar to our findings, methanol leaf extract of *Melastomastrum capitatum* (Vahl) A.Fern. and R.Fern was found to be composed of fatty acid-like 9-dodecanoic acid methyl ester, methyl tetradecanoate, hexadecanoic acid methyl ester, methyl stearate, oleic acid and a novel compound methyl-18-nonadecanoate [27]. The anticancer activity of *Arisaema flavum* (Forssk.) chloroform extract can be linked to the presence of these fatty acid esters, as it is suggested that plants having fatty acid esters in their extract are more potent bioactive agents [28,29]. Another study reports the phytochemical profiling of *Solena amplexicaulis* (Lam.) Gandhi, indicating the presence of linoleic acid esters and hexadecanoic acid methyl ester. These compounds were reported to have anti-inflammatory, cancer-protective and curative, hepatoprotective, antiarthritic and anti-coronary properties [30,31,32,33]. On the basis of our findings, it is worth mentioning that the mechanism of action of these compounds needs to be investigated.

## 5. Conclusions

The results of this study concluded that *Arisaema flavum* (Forssk.) crude extract as well as its fractions has significant anticancer activity against breast cancer cell lines. The systematic separation and purification techniques such as column chromatography, thin-layer chromatography, free drying, etc., led to the successful isolation of pure compounds. The detailed characterization of isolated compounds with FT-IR, GC and GC-MS resulted in the structure elucidation of isolated compounds. The isolated compounds indicated the presence of ester derivatives in rhizomes of this important medicinal plant with anticancer/cytotoxic activity against breast cancer cell lines. Further effects of these compounds on normal cell lines are needed to establish their selective toxic nature against cancer cells, which will be studied in future work.

## Figures and Tables

**Figure 1 molecules-27-07932-f001:**
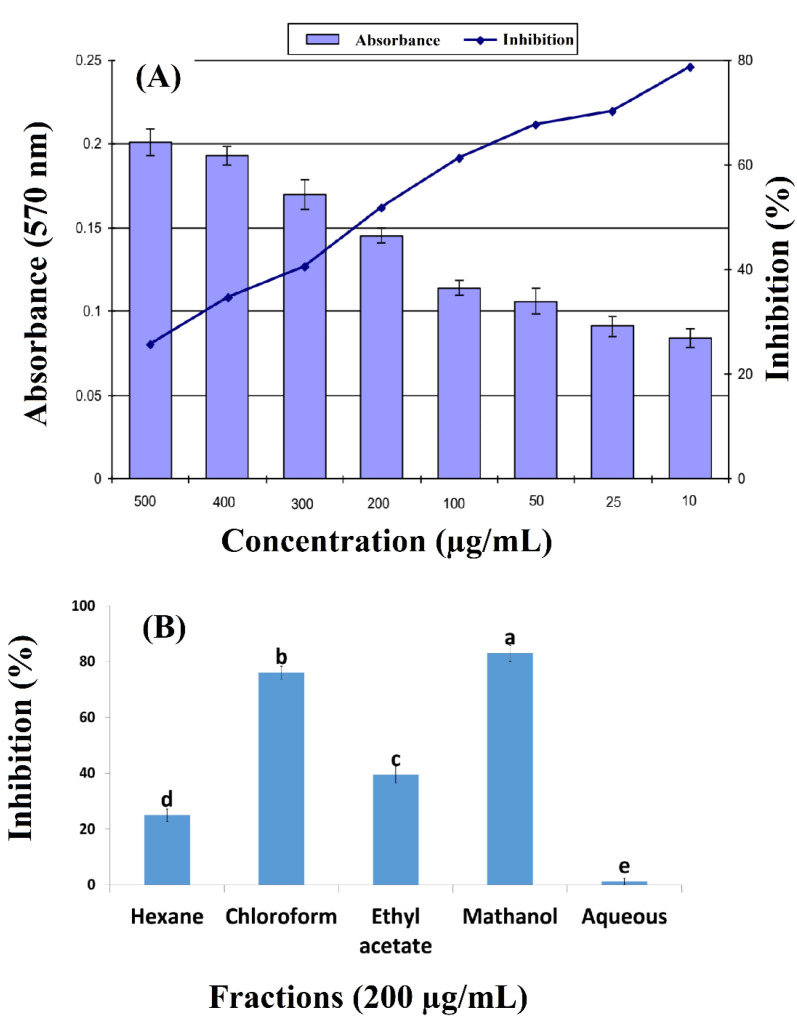
(**A**) Effect of concentration of *Arisaema flavum* (Forssk.) crude extracts on inhibition of breast cancer cell lines (MCF 7) growth and (**B**) effect of different fractions of *Arisaema flavum* (Forssk.) on inhibition of breast cancer cell lines (MCF 7) growth; a (methanol extract) and b (chloroform extract)) represents statistical significance at *p* value ≤ 0.05 as compared to c (ethyl acetate extract), d (n-hexane extract) and e (aqueous extract).

**Figure 2 molecules-27-07932-f002:**
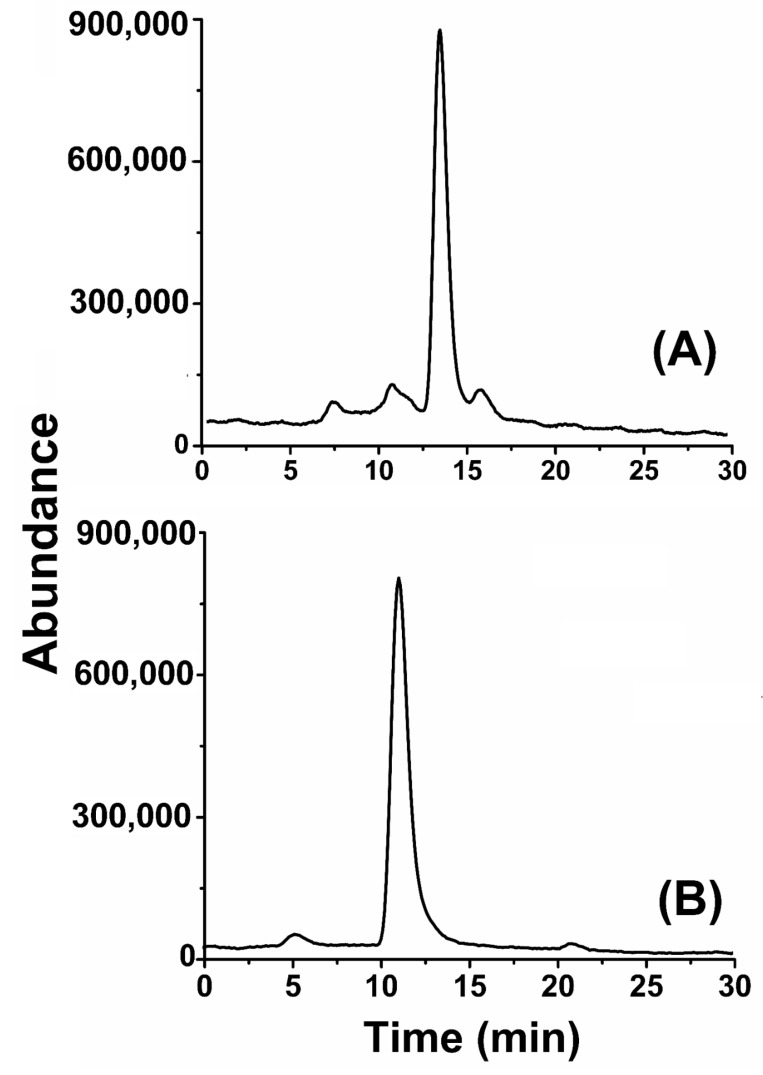
GC analysis of compound-I (**A**) and compound-II (**B**) isolated from *Arisaema flavum* (Forssk.).

**Figure 3 molecules-27-07932-f003:**
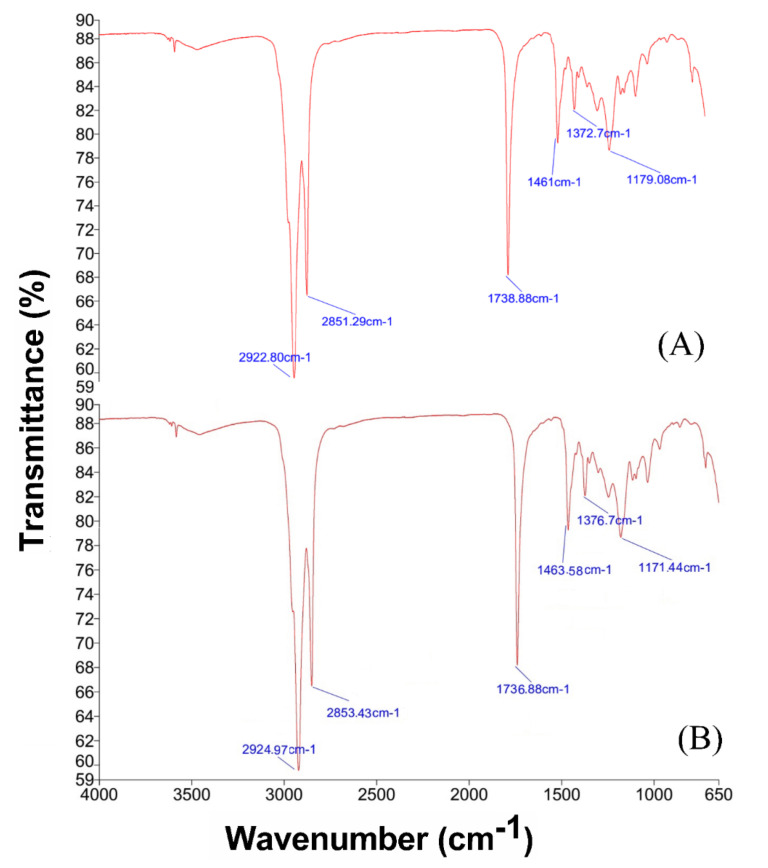
FTIR analysis of compound-I (**A**) and compound-II (**B**) isolated from *Arisaema flavum* (Forssk.).

**Figure 4 molecules-27-07932-f004:**
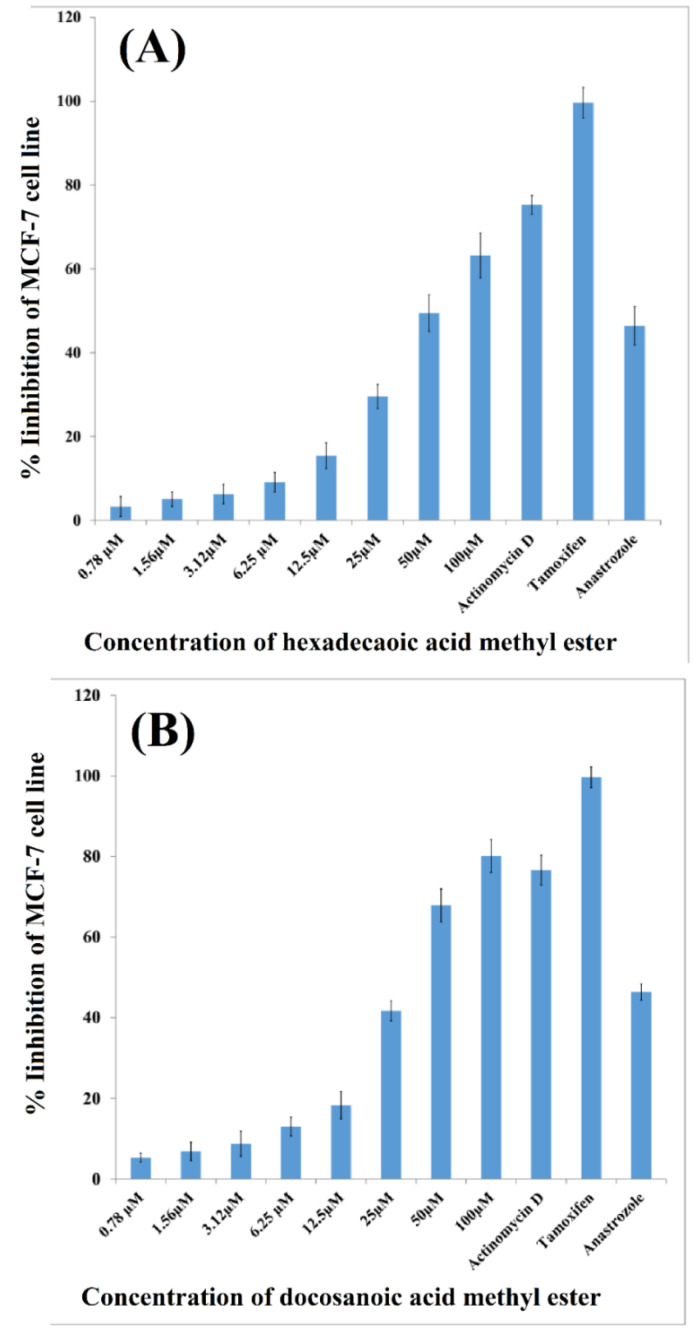
Anticancer activity of Hexadecanoic acid ethyl ester (**A**) and Docosanoic acid methyl ester (**B**) isolated from *Arisaema flavum* (Forssk.) against breast cancer cell line (MCF-7) after 48 h of exposure.

**Table 1 molecules-27-07932-t001:** Proton and Carbon NMR Summary of compound-I (Hexadecanoic acid ethyl ester) (A), and compound-II (5-Oxo-19 propyl-docosanoic acid methyl ester) (B) isolated from *Arisaema flavum* (Forssk.).

** ^1^ ** **H-NMR data of** **Compound-I (A)**			
**Carbon No**	** ^1^ ** **H(δ)**	** ^13^ ** **C(δ)**	**Multiplicity (DEPT)**
1		173.9	C = O
2	2.28(t)	34.4	CH_2_
3	1.60(d)	29.65	CH_2_
4–13	1.27(s)	29.66	(CH_2_)_10_
14	1.27(s)	29.70	CH_2_
15	1.29(m)	29.60	CH_2_
16	0.88(t)	29.15	CH_3_
17	4.11(dd)	22.70	-O-CH_2_
18	1.25(m)	22.70	CH_3_
** ^1^ ** **H-NMR data of** **Compound-II (B)**			
1		174.45	C = O
2		60.23	CH_2_
3	1.64(dd)	25.08	CH_2_
4	1.59(m)	34.48	CH_2_
5	2.30(dd)	188.45	C = O
6		34.21	CH_2_
7	2.30(dd)	25.05	CH_2_
8	1.59(m)	22.78	CH_2_
9–17	1.28(s)	29.77–29.44	(CH_2_)_10_
18	1.28(s)	32.01	CH_2_
19	1.28(s)	29.24	CH
20,21	1.59(m)	29.25	(CH_2_)_2_
22	1.28(s)	14.20	CH_3_
23,24	0.88(m)	29.35	(CH_2_)_2_
25	1.28(s)	14.20	CH_3_
26	0.88(m)	51.53	-O-CH_3_

A broad singlet was observed at *δ* 1.28, which integrated into 26 protons, indicating methylene groups.

## Data Availability

Not applicable.

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
