# Peer review of "Isolation, Characterization and Anticancer Activity of Two Bioactive Compounds from Arisaema flavum (Forssk.) Schott"

_molecules, 2022, doi:10.3390/molecules27227932_

Round 1

Reviewer 1 Report

The authors developed extracts and fractions of Arisaema flavum with two isolated compounds, which they tested for anticancer activity. 

General comments:

1. The MS should be proofread by a native English speaker, and typos should be corrected throughout MS.

2. the Results and Discussion section is confusingly structured as it stands. I would suggest splitting the sections for clarity. 

3. the figure captions should be better described to make it easier for the reader to understand what they are seeing in the figures.

Specific comments:

1. if the authors state in the abstract that about 80-85% of the world's population is dependent on herbal remedies, this should be supported by the references in the introduction.

2. line 171 - "Cell death can be either apoptotic or necrotic" is not true. There are other types of cell death and the authors cannot infer the type of cell death based on the tests used. 

3. line 172 - The IC50 value of 200 ug/ml does not correspond to the data in Figure 1. Can the authors explain this discrepancy? 

4. Figure 1A is missing the controls (untreated cells), which should be added to the figure. 

5. the description of the results of the spectroscopic analysis in section 3.2.1 is confusing and should be presented in a different format or included in the supplements. 

6. if the MTT assay was used for anticancer activity of the isolated compounds, this should be stated in the methods. What was the basis for selecting the concentrations of the positive controls? The title of the graph is still added. 

7. section 3.2 - Please explain why the chloroform fraction was used and not another, e.g. methanol, which has better inhibitory activity.

Author Response

Thank you so much for your valuable comments and suggestions. We have addressed your suggestions and point by point response to your comments is attached here. We hope we have made the changes according to your suggestions in the revised manuscript.  

Reviewer 2 Report

A manuscript titled “Isolation, Characterization and Anticancer Activity of two bioactive compounds from Arisaema flavum” provides insight into the significant anticancer activity of Arisaema flavum (Forssk.) Schott rhizome extract against breast cancer cell line. It also reveals the presence of ester derivatives in the rhizomes of this plant. The Manuscript is clear, concise, and provides a sufficient explanation of the aim, methods, and obtained results. I recommend it for publication, with minor changes needed (provided below).

Title and everywhere in the Manuscript – Arisaema flavum (Forssk.) Schott

Most of the references are out of date, more than 10 or 15 years old. Please provide newer ones where possible.

Abstract – crude rhizome extract or crude extract of rhizome

page 3, lines 106-108, and page 4, line 153 Why exactly those concentrations were chosen? Please provide an explanation

page 3, lines 130 and 133; page 4, line 152; pages 5 and 6 (lines 184, 186, and 196); fig 2 ... – compounds 1 and 2 or compounds I and II? Please make it uniform in the Manuscript

page 4, line 173 - Debregeasia salicifolia Rendle (DS) 

page 4, line 177 - Elaeis guineensis Jacq.

figure 1b – Please provide an explanation for a, b, c, d, and e

page 8, line 232 - Bauhinia purpurea L.

page10, line 289 - Melastomastrum capitatum (Vahl) A.Fern. & R.Fern.

page 10, line 300 - Solena amplexicaulis (Lam.) Gandhi

Author Response

(The authors gave the same response as above.)

Round 2

Reviewer 1 Report

The authors responded that Figure 1 had a labelling problem that should be corrected, but there is no difference from the previous version of the manuscript. In the discussion, the authors then explain: "Results indicated dose dependent response of breast cancer cell line with IC 50 value of 200 μg/mL. Similar dose dependent response of Debregeasia salicifolia Rendle (DS) was also reported against MCF 7 cell line [15]. Based on the IC 50 value of crude extract, 200 μg/mL concentration of fractions were selected for further processing."

Figure 1 clearly does not show IC 50 values at 200 μg/mL so this concentration does not seem to be justified for further experiments. Again, how can the authors explain this? 

The manuscript does not seem to have been proofread, as there are still many typos throughout the text.

Author Response

Response to reviewer 2 comments (Round 2)

The authors responded that Figure 1 had a labelling problem that should be corrected, but there is no difference from the previous version of the manuscript.

Response: The corrected Fig 1 was added in the revised manuscript with track changes but the changes were not saved. Corrected Fig 1 is added again in the revised manuscript.

In the discussion, the authors then explain: "Results indicated dose dependent response of breast cancer cell line with IC 50 value of 200 μg/mL. Similar dose dependent response of Debregeasia salicifolia Rendle (DS) was also reported against MCF 7 cell line [15]. Based on the IC 50 value of crude extract, 200 μg/mL concentration of fractions were selected for further processing."

Figure 1 clearly does not show IC 50 values at 200 μg/mL so this concentration does not seem to be justified for further experiments. Again, how can the authors explain this? 

Response: Figure 1 is corrected in the revised manuscript. Secondly, effect of concentration of Arisaema flavum (Forssk) crude extract on inhibition of breast cancer cell lines (MCF 7) growth is shown in Supplementary Fig 8. Suppl Fig 8 clearly shows that 50% inhibition of breast cancer cell lines (MCF 7) growth occurred at 200 μg/mL.

The manuscript does not seem to have been proofread, as there are still many typos throughout the text.

Response: The manuscript was proofread and all the typos are corrected.

Round 3

Reviewer 1 Report

The manuscript can be considered for publication.